# Pharmacologically induced amnesia for learned fear is time and sleep dependent

Merel Kindt[1,2] & Marieke Soeter[1,3]

The discovery in animal research that fear memories may change upon retrieval has sparked a wave of interest into whether this phenomenon of reconsolidation also occurs in humans. The critical conditions under which memory reconsolidation can be observed and targeted in humans, however, remain elusive. Here we report that blocking beta-adrenergic receptors in the brain, either before or after reactivation, effectively neutralizes the expression of fear memory. We show a specific time-window during which beta-adrenergic receptors are involved in the reconsolidation of fear memory. Finally, we observe intact fear memory expression 12 h after reactivation and amnesic drug intake when the retention test takes place during the same day as the intervention, but post-reactivation amnesia after a night of sleep (12 h or 24 h later). We conclude that memory reconsolidation is not simply time-dependent, but that sleep is a final and necessary link to fundamentally change the fear memory engram.

[1] Department of Clinical Psychology, University of Amsterdam, Nieuwe Achtergracht 129B, 1018 WS Amsterdam, The Netherlands. [2] Amsterdam Brain and Cognition Center, Nieuwe Achtergracht 129B, 1018 WS Amsterdam, The Netherlands. [3] Microbiology and Systems Biology, TNO, Utrechtseweg 48, 3704 HE Zeist, The Netherlands. Correspondence and requests for materials should be addressed to M.K. (email: m.kindt@uva.nl)

Observations of post-reactivation amnesia for learned fear have drastically changed the neuroscience literature on memory by generating a novel and influential conceptual framework, known as memory reconsolidation[1–5]. It refers to the idea that reactivation can induce a transient, labile state in a previously stable memory, during which the memory is sensitive to disruption or modification, necessitating a time-dependent process of restablization in order to persist. Gene transcription, RNA translation, and protein synthesis are necessary for reconsolidation and these neurobiological processes enable to fundamentally change a fear memory engram[2, 5]. Post-reactivation amnesia for learned fear has now been demonstrated in a range of model organisms, from fish, to crabs, to rats[5]. These findings have provided a strong impetus for research into whether this phenomenon of post-reactivation amnesia can be observed in humans as well[6]. Treatments for disorders of emotional memory generally involve multiple or prolonged sessions in which the fear gradually subsides[7]. A single intervention that instantaneously neutralizes the emotional impact of fearful memories would signify a true paradigm shift in the practice of psychotherapy. An abrupt reduction in fear responding by a single amnesic drug administered upon a very brief memory reactivation has been repeatedly demonstrated in healthy participants, and more recently in a clinical sample as well[8–17]. Notwithstanding these promising findings, a limitation in human memory research is that the underlying neurobiological processes of memory reconsolidation can neither be directly observed nor locally targeted. Hence, the critical conditions for harnessing memory reconsolidation in humans are still largely unknown.

Animal and human studies have demonstrated that prediction error—operationalized as a match-mismatch between a past and present experience—is necessary to destabilize a previously formed fear memory[15–19]. If a match-mismatch experience is induced by memory reactivation in humans, blocking beta-adrenergic receptors (β-ARs) erases the later mnemonic output for learned fear[15–17]. Animal research on memory consolidation provides direct in vivo evidence that under weak training conditions both activation of β-ARs in the lateral amygdala and Hebbian plasticity mechanisms (neuronal depolarization) are necessary to trigger late long-term potentiation (L-LTP)[20, 21]. Even though β-ARs are not required for the induction of synaptic potentiation, they can modify the ability of synapses to undergo plasticity by lowering the threshold for induction of LTP or by extending the time-scales well beyond normal synaptic transmission[21–23]. Memory encoding and tagging (early-LTP) occur in real time triggered by the event or stimulus to be remembered, but the eventual persistence of this trace depends upon the capture of plasticity-related proteins whose synthesis can be triggered before, during, or after memory encoding[23–25]. If plasticity-related proteins are not available during the window of synaptic tagging (e-LTP), the receptive synapses will fade away preventing the formation of long-term memory (i.e., the synaptic tag decays in <3 h)[26, 27]. In line with the alleged functional role of reconsolidation to update memories to an ever-changing environment[3], we hypothesize that prediction error (i.e., match-mismatch) during memory reactivation has a twofold function: (a) it sets a synaptic re-learning tag by destabilizing the glutamatergic NMDA receptors underlying the fear memory engram[5, 28], and (b) it increases noradrenergic and dopaminergic activity acting on centrally located β-adrenergic and dopaminergic D1/D5 receptors, respectively[22, 29]. For β-ARs, it has been demonstrated that they have an essential role in PRP synthesis required for memory consolidation and reconsolidation, either via the canonical G proteins/cAMP/PKA/CREB pathway[30–33], or

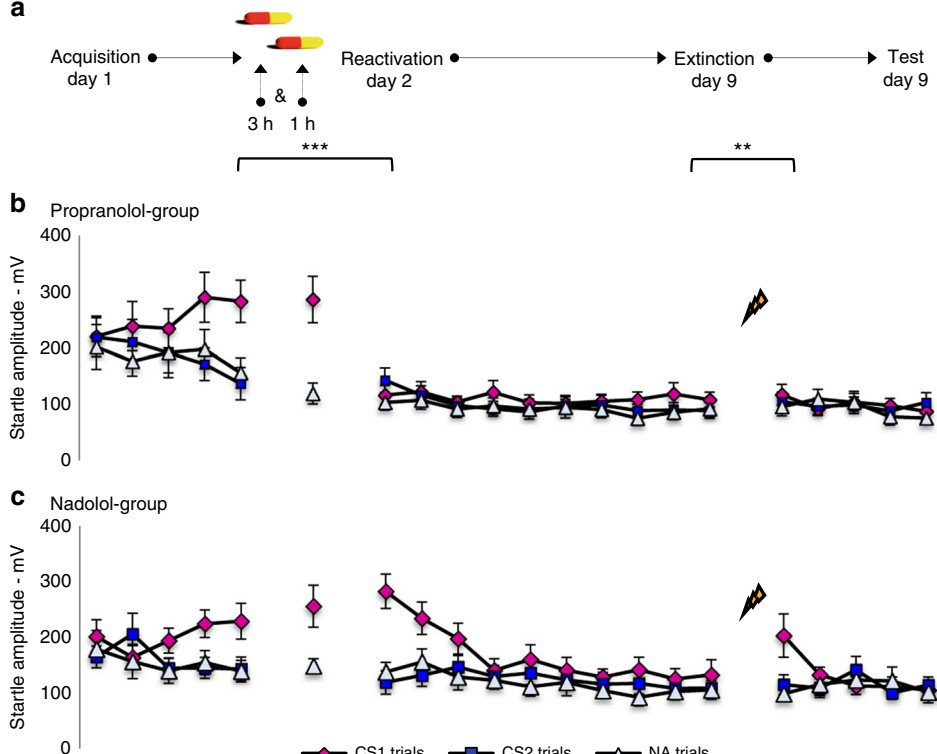

**Fig. 1** Blocking central β-ARs disrupts fear memory reconsolidation. **a** Experimental design and timeline for the propranolol group and nadolol group. **b** Mean startle potentiation to the fear conditioned stimulus (CS1—pink diamonds), the control stimulus (CS2—blue squares), and the noise alone (NA—gray triangles) trials during acquisition as well as reactivation and test for the propranolol group and **c** the nadolol group. Error bars represent s.e.m. *P < 0.05, **P < 0.01, ***P < 0.001, within-between repeated measures ANOVA

via other downstream signaling pathways such as ERK/MAPK[22, 23, 34].

Building on these neurobiological insights from the animal literature, we inferred the following hypotheses with the aim to reveal critical conditions to target and observe memory reconsolidation in humans. If blocking the β-ARs disrupts the production of plasticity-related proteins independent from setting the re-learning tag, amnesia should be observed for the reactivated memory trace when a centrally active β-blocker is administered either before or after memory reactivation (experiment 1 to 3). Given that the receptive state of the synapses is transient, the β-AR blocker should be given within a specific time-window upon memory reactivation, or else the plasticity-related proteins will be produced and captured by the re-learning tag keeping the memory alive (experiment 2). As the receptive—or so-called tagged state—of the synapses is time-limited, the mnemonic output of the reconsolidation process should not coincide with the time frame of the synaptic tagging. On the contrary, the observation in animals of intact freezing several hours (4 h) after the amnesic drug administration (i.e., anisomycin, a protein synthesis blocker) indicates that the structural changes underlying the expression of the previously formed fear memory may remain functional for at least 4 h[35]. The temporal dissociation between the manipulation and expression of memory can be further explained by the specific and dissociable glutamate receptors mediating either the expression, destabilization, or restabilization of memory[28, 36]. If the β-ARs are necessary for the time-dependent restabilization process, the mnemonic output should be intact in the first hours after administering the amnesic drug (β-blocker), while post-reactivation amnesia should only be observed several hours later (experiment 3). Here, we report that using a single pill of propranolol (40 mg)—administered in conjunction with memory reactivation—to block β-ARs in humans effectively neutralizes associative fear memory. However, this post-retrieval amnesia for learned fear only became apparent when (a) the amnesic drug passed the blood–brain barrier, thereby targeting the centrally located β-ARs; (b) the amnesic drug was given within a specific time-window that spanned the time before, during, and after memory reactivation; and (c) the participants were tested 12 or 24 h later and that this delay included a night of sleep.

## Results

**Blocking central β-ARs disrupts fear memory reconsolidation.** In a cued fear-conditioning paradigm, the β-blocker propranolol is supposed to act specifically on β-ARs in the basolateral amygdala, a brain area essential for the formation, consolidation and expression of fear memories[34, 37, 38]. Given that propranolol HCl blocks β-ARs located both centrally in the brain and peripherally in the heart[39], it remains unclear whether this drug actually targets the synaptic plasticity of fear memories. If the administration of propranolol HCl specifically disrupts the process of memory restabilization and not memory destabilization, then a β-AR blocker that (a) passes the blood–brain barrier and (b) is administered before memory reactivation with its peak during reactivation should result in subsequent amnesia. To differentiate between the sites of β-adrenergic action[40, 41], we compared the effect of the lipophilic β-AR antagonist propranolol HCl (40 mg), that easily crosses the blood–brain barrier, with the more peripherally active hydrophilic (lipophobic) β-AR antagonist nadolol (40 mg), a water-soluble drug that crosses the blood–brain barrier to a much lesser extent than propranolol[42].

Participants (n = 30–13 men) underwent a differential fear-conditioning procedure that consisted of different phases across three days—see Fig. 1a. During acquisition a fear-relevant conditioned stimulus (CS1) was repeatedly paired with an electric stimulus whereas a fear-relevant conditioned stimulus (CS2) was not. A day later participants got 40 mg of either propranolol or nadolol. Given the disparity in peak-plasma concentrations between the drugs[42, 43], propranolol ($t_{max}$ = 1–2 h) was provided 1 h and nadolol ($t_{max}$ = 3–4 h) 3 h before memory reactivation (MR). To control not only for the drug per se but also for the timing of drug administration, both drugs were double-blind placebo-controlled, such that all participants were administered two pills. Half of the participants (n = 15) firstly received one placebo pill (3 h before MR) followed by one pill of propranolol (1 h before MR) and the other half of the participants (n = 15) firstly received one pill of nadolol (3 h before MR) followed by one placebo pill (1 h before MR). During MR, a single unreinforced CS1 (CS1-R, referring to reactivation) was presented. To guarantee that the drugs were washed out during the retention tests, memory was tested 1 week later (day 9) instead of the usual 24 h[8, 35], given the longer half-time value of nadolol ($t_{1/2}$ = ±24 h) compared to propranolol ($t_{1/2}$ = 5 h). Another reason to postpone the retention tests was to critically examine whether there are still savings of the original fear memory. It is clear that we cannot proof the absence of fear memory, but in order to enhance the likelihood of fear memory expression, we first tested for spontaneous recovery (i.e., retention test at day 9), followed by a behavioral provocation (i.e., reminder shocks) to test for reinstatement of the original fear learning. We measured conditioned fear responding as potentiation of the eye-blink startle reflex to a loud noise by electromyography (EMG) of the right orbicularis oculi muscle. The potentiation of the startle reflex reflects the negative valence of the fear conditioned stimulus (CS1) following its pairing with an aversive unconditioned stimulus (i.e., electric stimulus, US)[44]. In human fear-conditioning research, the fear-potentiated startle is typically inferred from the differential startle responding to the reinforced stimulus (CS1–>US) vs. the nonreinforced stimulus (CS2–>no-US) or noise alone probes (NA). US-expectancy ratings were used to assess the anticipation of threat (see Supplementary Figs. 1, 2 and 3 for all of these data).

Analyses of variances showed fear conditioning in both groups, as evidenced by a significant increase of the differential startle responding (CS1 vs. CS2) from trial 1 to trial 5 [stimulus × trial—$F_{1,28}$ = 13.08, P = 0.001, $\eta_p^2$ = 0.32]—see Fig. 1. Although a trend in group differences emerged during acquisition [stimulus × trial × group—$Fs_{1,28}$ < 3.17, P = 0.086], we observed comparable levels of startle potentiation during MR [CS1-R vs. NA—stimulus—$F_{1,28}$ = 38.84, P < 0.001, $\eta_p^2$ = 0.58—stimulus × group—$F_{1,28}$ < 1] and the fear responding remained stable from the last trial of acquisition to MR [CS1 vs. NA—stimulus × trial —$F_{1,28}$ < 3.59, P = 0.068—stimulus × trial × group—$F_{1,28}$ < 1]. The two drugs also exerted a similar physiological effect [systolic and diastolic BP, heart rate—moment × group—$Fs_{1,28}$ < 1]. However, the propranolol group showed a significant decrease in startle fear responses to the CS1 from the last trial of acquisition to the first extinction trial 1 week later as compared to nadolol group [stimulus × trial × group—$F_{1,28}$ = 34.69, P < 0.001, $\eta_p^2$ = 0.55]. Planned comparisons indeed showed that the propranolol manipulation strongly reduced the emotional expression of the CS1 memory [stimulus × trial—$F_{1,14}$ = 31.60, P < 0.001, $\eta_p^2$ = 0.69], whereas the differential startle responding even increased in the nadolol group [$F_{1,14}$ = 6.99, P < 0.05, $\eta_p^2$ = 0.33]. As a result, the extinction learning process also differed between the two groups [stimulus × trial × group—$F_{1,28}$ = 32.13, P < 0.001, $\eta_p^2$ = 0.53]. Fear responding to the CS1 significantly decreased in the nadolol group [trial—$F_{1,14}$ = 22.81, P < 0.001, $\eta_p^2$ = 0.62]. Yet no differential change in fear responding was observed in the propranolol group [$F_{1,14}$ < 2.94], as the learned fear did not return

during the extinction phase. Furthermore, a recovery of fear was observed from the last extinction trial to the first test trial in the nadolol group [stimulus × trial—$F_{1,14} = 12.33$, $P < 0.01$, $\eta_{\mathrm{p}}^2 = 0.47$], but not in the propranolol group [stimulus × trial—$F_{1,14} < 1$—stimulus × trial × group—$F_{1,28} = 10.10$, $P < 0.01$, $\eta_{\mathrm{p}}^2 = 0.27$].

Overall these data support the hypothesis that the fear-erasing effect of propranolol is centrally mediated, as the peripheral β-AR antagonist nadolol administered before memory reactivation shows no effect on subsequent fear responding[40, 41]. Crucially, the administration of propranolol 1 h before the reminder trial did not affect the mnemonic output during MR (day 2) (i.e., no decrease in fear responding from the end of acquisition to MR). Hence, the observation that the administration of propranolol 1 h before the reminder trial (MR) on day 2 affected the startle fear response 7 days later (i.e., when the drug is entirely washed out) is in line with the hypothesis that β-ARs are specifically involved in the process of memory restabilization.

**β-ARs are critical within a specific time-window.** Administration of an amnestic agent may permanently diminish the expression of fear memory if presented within the reconsolidation window: the period wherein the synapses of the memory engram are in the receptive state. The synaptic tagging and capture hypothesis predicts a window of about 3 h during which PRPs can be captured for memory consolidation. In several of our previous studies on memory reconsolidation we administered the β-AR antagonist propranolol directly following memory reactivation and observed amnesia 24 h later[11, 12, 16, 17]. In addition, systemic administration of propranolol in rats triggered amnesia when the injections were made up to 2 h after the memory reactivation[45].

In view of the limited lifespan of the tagged state[24] and the pharmacokinetic signature of propranolol ($t_{\max} = 1$–2 h; $t_{1/2} = 5$ h)[43], we hypothesized that β-ARs are implicated within a specific time-window in the late phase of memory reconsolidation. Contrary to animal studies—in which drugs can be directly inserted into the brain—we are bound to administer the β-AR blocker propranolol in a systemic fashion. However, by exploiting the bioavailability of propranolol, it is also possible to demarcate the critical time-window during which β-ARs are involved in the reprocessing of fear memories in humans. Participants were subjected to a mixed within-between-subjects fear-conditioning procedure, allowing to test the effect of disrupting memory reconsolidation within and between-subjects. During acquisition (day 1), we repeatedly paired two categorically distinct stimuli (CS1–>US and CS2–>US) with an aversive electric stimulus (US). A third stimulus was presented without the US (CS3–>no-US). On day 2, only one of the two fear associations was reactivated (referred to as CS1-R) and subsequently manipulated by the systemic administration of propranolol HCl, while the other fear association was not reactivated (referred to as CS2)—see Fig. 2a. A series of pilot cases with varying timing between MR and propranolol administration revealed that the β-AR window seemed to close somewhere between 1 to 2 h post-reactivation pill intake—see Supplementary Fig. 4 for the pilot data. On the basis of a G*Power analysis ($f = 0.35$, $\alpha = 0.05$, $1$–$\beta = 0.95$), 20 participants (9 men) received single blind an oral dose of 40 mg of propranolol exactly 1 h ($n = 10$) or 2 h ($n = 10$) after memory reactivation.

Fear learning was observed in both the pill_1h and pill_2h group [i.e., simple contrasts: CS1 vs. CS3 and CS2 vs. CS3—stimulus × trial—$F_{1,18} = 41.16$, $P < 0.001$, $\eta_{\mathrm{p}}^2 = 0.70$ and

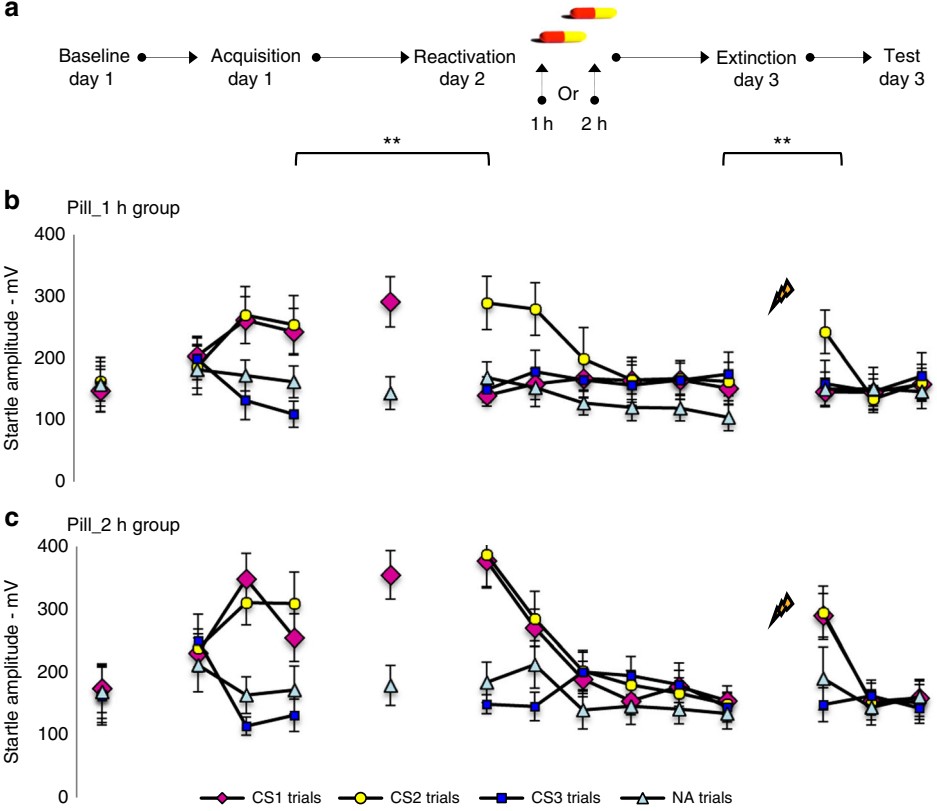

**Fig. 2** β-ARs are critical for memory reconsolidation within a specific time-window. **a** Experimental design and timeline for the pill_1h and pill_2h groups. **b** Mean startle potentiation to the fear conditioned stimuli (CS1—pink diamonds; CS2—yellow dots ), the control stimulus (CS3—blue squares), and the noise alone (NA—gray triangles) trials during acquisition as well as reactivation and test for the pill_1h group and **c** the pill_2 h group. Error bars represent s.e.m. *$P < 0.05$, **$P < 0.01$, ***$P < 0.001$, within-between repeated measures ANOVA

$F_{1,18} = 22.61$, $P < 0.001$, $\eta_p^2 = 0.56$—stimulus × trial × group—$Fs_{1,18} < 1$]—see Fig. 2. We observed no difference in responding to baseline testing [CS1 vs. CS2 vs. CS3—stimulus × group—$Fs_{2,16} < 1$], which shows that the fear relevancy of the pictures was not affecting the startle responding in the absence of associative learning[10, 11]. Fear responses to the reinforced pictures were also equally acquired [CS1 vs. CS2—stimulus × trial—stimulus × trial × group—$Fs_{1,18} < 1.39$]. Furthermore, the two groups expressed similar levels of startle potentiation during MR [CS1-R vs. NA—stimulus—$F_{1,18} = 68.05$, $P < 0.001$, $\eta_p^2 = 0.79$—stimulus × group—$F_{1,18} < 1$] and responding to the non-reactivated CS2 did not change from acquisition to extinction learning 48 h later [CS2 vs. CS3—stimulus × trial—stimulus × trial × group—$Fs_{1,18} < 1$]. Conversely, in the pill_1h group, but not in the pill_2h group, we observed a significant decrease in fear responses to the reactivated CS1 from the last trial of acquisition to the first extinction trial 48 h later [CS1 vs. CS3—stimulus × trial × group—$F_{1,18} = 14.86$, $P = 0.001$, $\eta_p^2 = 0.45$]. Planned comparisons showed that the propranolol manipulation indeed reduced the emotional expression of the reactivated fear memory in the pill_1h group [CS1 vs. CS3—$F_{1,9} = 13.15$, $P = 0.01$, $\eta_p^2 = 0.59$], whereas the differential startle responding even increased somewhat in the pill_2h group [CS1 vs. CS3—$F_{1,9} < 4.36$, $P = 0.066$]. We also observed diminished fear responses to the reactivated CS1 contrary to the non-reactivated CS2 in the pill_1h group [CS1 vs. CS2—stimulus × trial—$F_{1,9} = 4.83$, $P = 0.05$, $\eta_p^2 = 0.35$], but not in the pill_2 h group [CS1 vs. CS2—stimulus × trial—$F_{1,9} < 1$—trial × group—$F_{1,18} = 6.54$, $P < 0.05$, $\eta_p^2 = 0.27$]. Consequently, fear responding to the reactivated CS1 significantly decreased during extinction training in the pill_2h group [CS1 vs.

CS3—stimulus × trial—$F_{1,9} = 23.43$, $P = 0.001$, $\eta_p^2 = 0.72$] as compared to the pill_1h group [CS1 vs. CS3—stimulus × trial—$F_{1,9} < 1$—stimulus × trial × group—$F_{1,18} = 16.53$, $P = 0.001$, $\eta_p^2 = 0.48$]. However, in both groups, startle responses to the non-reactivated CS2 significantly decreased from the first to the last extinction trial [CS2 vs. CS3—stimulus × trial—$F_{1,18} = 26.34$, $P < 0.001$, $\eta_p^2 = 0.59$—stimulus × trial × group—$F_{1,18} < 1.16$]. Furthermore, startle responses to the non-reactivated CS2 significantly increased following the reminder shocks [CS2 vs. CS3—stimulus × trial—$F_{1,18} = 25.22$, $P < 0.001$, $\eta_p^2 = 0.58$—stimulus × trial × group—$F_{1,18} < 1$]. The reminder shocks did not uncover any fear responding to the CS1 in the pill_1h group [CS1 vs. CS3—$F_{1,9} < 1$], contrary to the pill_2h group [CS1 vs. CS3—stimulus × trial—$F_{1,9} = 25.39$, $P = 0.001$, $\eta_p^2 = 0.74$—stimulus × trial × group—$F_{1,18} = 12.08$, $P < 0.01$, $\eta_p^2 = 0.40$]. We indeed found a significant reinstatement effect to the non-reactivated CS2 as compared to the reactivated CS1 in the pill_1h group [CS1 vs. CS2—stimulus × trial—$F_{1,9} = 16.91$, $P < 0.01$, $\eta_p^2 = 0.65$], but no difference in reinstatement was found for the CS1 and CS2 in the pill_2h group [CS1 vs. CS2—stimulus × trial—$F_{1,9} < 1$—stimulus × trial × group—$F_{1,18} = 5.79$, $P < 0.05$, $\eta_p^2 = 0.24$].

Here we demonstrate a significant role for β-ARs in the modulation of fear memory expression, but only if they are targeted during a specific time-window. In view of the pharmacokinetics of propranolol HCl ($t_{max} = 1$–2 h; $t_{1/2} = 5$ h) and the observation that drug administration either 1 h before (exp_1) or 1 h after (exp_2) MR affected the emotional expression of fear memory, we infer that during the first 1–2 h post-reactivation, β-ARs are not required for memory reconsolidation. The observation that drug administration 2 h after reactivation

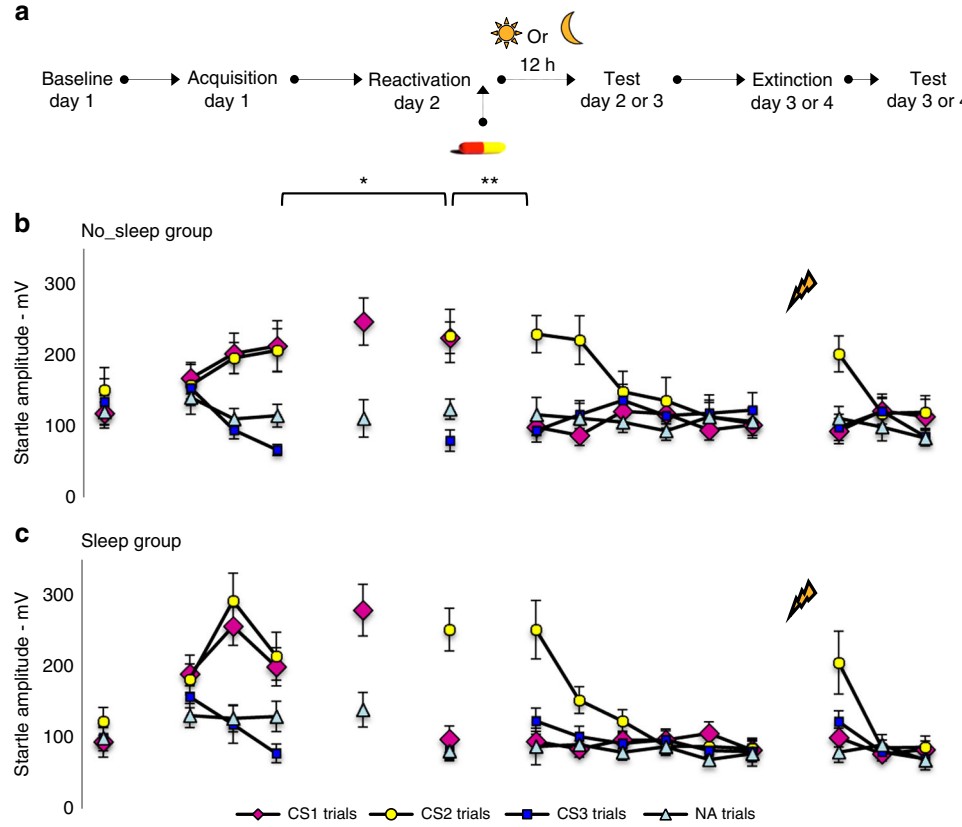

**Fig. 3** Post-reactivation amnesia requires sleep. **a** Experimental design and timeline for the no_sleep and sleep groups. **b** Mean startle potentiation to the fear conditioned stimulus (CS1—pink diamonds; CS2—yellow dots), the control stimulus (CS3—blue squares), and the noise alone (NA—gray triangles) trials during acquisition as well as reactivation and test for the no_sleep group and **c** the sleep group. Error bars represent s.e.m. *$P < 0.05$, **$P < 0.01$, ***$P < 0.001$, within-between repeated measures ANOVA

did not affect the later expression of fear memory suggests that β-ARs are involved in memory reconsolidation during a very small time-window: between 2 and 3 h post-reactivation (Supplementary Fig. 5).

**Post-reactivation amnesia requires sleep.** To directly test whether the β-AR blocker propranolol actually disrupts the process of memory reconsolidation, the expression of fear memory should be intact in the first hours following reactivation, given that the required protein synthesis for reconsolidation takes at least several hours[35]. Although dozens of human memory studies are conceptualized in the memory reconsolidation framework[17, 46], a litmus test for reconsolidation is to show a desynchronization between initially intact memory expression following the intervention and subsequent amnesia. A series of pilot cases with varying timing between reactivation and test disclosed that the memory was still intact when tested 1 h, 5 h and even 12 h after MR on the same day, but seriously affected 24 later (Supplementary Fig. 6). Given that sleep is involved in the consolidation of new memories[47, 48], we hypothesized that it might also be crucial for the reconsolidation of fear memories[49]. For addressing this question participants ($n = 20–12$ men) were again exposed to a mixed within-between-subject fear-conditioning design and were tested exactly 12 h after MR: either following a day of wakefulness or a night of sleep (see Fig. 3a for the design).

Analyses of variances showed fear learning in both the no_sleep and sleep group by a significant increase of differential startle responding from trial 1 to trial 3 [i.e., simple contrasts: CS1 vs. CS3 and CS2 vs. CS3—stimulus × trial—$F_{1,18} = 17.23$, $P = 0.001$, $\eta_p^2 = 0.49$ and $F_{1,18} = 36.00$, $P < 0.001$, $\eta_p^2 = 0.67$], without a group difference [stimulus × trial × group—$Fs_{1,18} < 1$]—see Fig. 3. We observed no difference in responding to the baseline measurement [CS1 vs. CS2 vs. CS3—stimulus × group—$F_{2,16} < 1$] and fear responses to the reinforced pictures were equally acquired [CS1 vs. CS2—stimulus × trial—stimulus × trial × group—$Fs_{1,18} < 1$]. Moreover, we observed comparable levels of startle potentiation during MR on day 2 [CS1-R vs. NA—stimulus $F_{1,17} = 77.68$, $P < 0.001$, $\eta_p^2 = 0.82$—stimulus × group—$F_{1,17} < 1$]. However, in the sleep group, but not in the no_sleep group, the administration of propranolol significantly decreased startle fear responses to the reactivated CS1 from the last trial of acquisition to the first retention test 12 h after MR [CS1 vs. CS3—CS1 vs. CS2—stimulus × trial × group—$F_{1,17} = 8.28$, $P = 0.01$, $\eta_p^2 = 0.33$—$F_{1,17} = 5.07$, $P < 0.05$, $\eta_p^2 = 0.23$, respectively]. Planned comparisons showed that the drug strongly reduced the emotional expression of the reactivated CS1 memory in the sleep group [CS1 vs. CS3—CS1 vs. CS2—$F_{1,8} = 24.05$, $P = 0.001$, $\eta_p^2 = 0.75$—$F_{1,8} = 11.63$ $P < 0.01$, $\eta_p^2 = 0.59$, respectively], but not in the no_sleep group [CS1 vs. CS3—CS1 vs. CS2—$Fs_{1,9} < 1$]. Conversely, responding to the non-reactivated CS2 remained stable from acquisition to the first retention test 12 h after reactivation in both groups [CS2 vs. CS3—stimulus x trial—stimulus × trial × group—$Fs_{1,17} < 1$].

Given that the fear-potentiated startle response to the reactivated CS1 was eliminated on the first retention test in the sleep group [CS1 vs. CS3—$t_9 < 1$—CS1 vs. CS2—$t_8 = 5.01$, $P < 0.001$, two-tailed], we observed a significant difference between groups in fear responding to the CS1 from this test trial to the first extinction trial 24 h later [CS1 vs. CS3—CS1 vs. CS2—stimulus × trial × group—$F_{1,17} = 9.16$, $P < 0.01$, $\eta_p^2 = 0.35$—$F_{1,17} = 3.77$, $P = 0.069$, $\eta_p^2 = 0.18$, respectively]. Interestingly, the fear response to the reactivated CS1 also significantly decreased in the no_sleep group at the second retention test after a night of sleep (i.e., from day 2 to day 3) [CS1 vs. CS3—CS1 vs. CS2—$F_{1,9} = 42.07$, $P < 0.001$, $\eta_p^2 = 0.82$—$F_{1,9} = 30.48$,

$P < 0.001$, $\eta_p^2 = 0.77$, respectively]. We observed no differential change in fear responding in the sleep group at the second retention test (i.e., from day 3 to day 4), indicating that the neutralization of the fear memory remained stable [CS1 vs. CS3—CS1 vs. CS2—$Fs_{1,8} < 1$]. Fear responding to the non-reactivated CS2 again remained stable in both groups from this retention test trial to the first trial of extinction learning 24 h later [CS2 vs. CS3—stimulus × trial—stimulus × trial × group—$Fs_{1,17} < 1$]. Yet in the sleep as well as no_sleep group we observed a significant decrease in responses to the non-reactivated CS2 from the first trial of extinction learning to the last extinction trial [CS2 vs. CS3—stimulus × trial—$F_{1,17} = 52.27$, $P < 0.001$, $\eta_p^2 = 0.76$—stimulus × trial × group—$F_{1,17} < 1.05$], but we no longer observed a differential change in startle fear responding to the reactivated CS1 [CS1 vs. CS3—stimulus × trial × group—$F_{1,17} < 1.86$]. Furthermore, although the reminder shocks did not uncover any fear responding to the reactivated CS1 in the two groups [CS1 vs. CS3—stimulus × trial—stimulus × trial × group—$Fs_{1,17} < 1$], fear responding to the non-reactivated CS2 significantly increased from the last trial of extinction to the first trial at test in both groups [CS2 vs. CS3—CS1 vs. CS2—stimulus × trial—$F_{1,17} = 16.33$, $P = 0.001$, $\eta_p^2 = 0.49$—$F_{1,18} = 8.12$, $P < 0.05$, $\eta_p^2 = 0.31$—stimulus × trial × group—$Fs_{1,17} < 1$, respectively].

To summarize, in both groups post-reactivation amnesia became only apparent after a night of sleep (12 h or 24 h later), while the fear expression was still intact when retention took place at the same day of the intervention—even after 12 h—but without a night of sleep in between (Fig. 3). Although sleep was not registered in our study, the finding that post-reactivation amnesia emerged the following day in both groups, suggests that sleep is critically involved in reconsolidation-induced forgetting. The notion that sleep is involved in memory reconsolidation is already well-established, given the repeated observations of memory-reactivation improving effects following sleep[50–52]. Sleep is well-known to strengthen and integrate new memories into pre-existing networks, but there is no experimental evidence that sleep is actually required to observe post-reactivation amnesia for learned fear. Here we show a crucial role for sleep to observe the absence of memory expression triggered by the earlier reconsolidation intervention.

## Discussion

We leveraged temporally precise, memory-specific manipulations to test the critical conditions to target and observe β-AR induced post-reactivation amnesia for learned fear in humans. We found that a β-AR antagonist effectively neutralized the fear-potentiated startle response, but exclusively when the drug passed the blood–brain barrier (i.e., propranolol but not nadolol), and when given within a specific time-window upon memory reactivation. The β-AR antagonist propranolol produced subsequent amnesia for learned fear when it was administered 1 h (exp_1) before memory reactivation and also when it was administered after memory reactivation, either directly following the reminder trial (exp_3) or 1 h later (exp_2). The fear memory was however impervious to the amnestic effects of propranolol when administered 2 h after memory reactivation (exp_2). Most remarkably, we observed intact fear memory expression when the participants were tested 12 h after memory reactivation and amnesic drug intake at the same day, whereas post-retrieval amnesia only emerged after a night of sleep (either 12 h or 24 h later).

Our findings suggest that the β-ARs are specifically involved in the process of memory restabilization during a critical time-window, and that this window is relatively small. In view of (a) the different timings of drug administration, (b) the pharmacokinetic signature of propranolol ($t_{max} = 1–2$ h; $t_{1/2} = 5$ h) and (c)

the presence or absence of post-reactivation amnesia, we postulate a late β-AR involvement (2–3 h post-reactivation) in the reconsolidation of fear memory. If early β-AR activity before the 2–3 h time-window had been responsible for the post-reactivation amnesia, then the administration of propranolol 1 h following memory reactivation would have missed this window. An early β-AR activity in this group would then have triggered the neurobiological cascade of cAMP/PKA/CREB[30–33] resulting in memory reconsolidation of the destabilized fear memory. At a behavioral level, these neurobiological processes would have resulted in the persistence of fear memory expression instead of post-reactivation amnesia. In a similar vein, if a peak of β-AR activity after 3 h had been critical for memory reconsolidation, the administration of propranolol 1 h before memory reactivation would probably have missed this window, whereas in this group the administration of propranolol also disrupted the process of memory reconsolidation.

Although the current findings suggest a narrow and delayed time-window (2–3 h post-reactivation) for β-AR activity, a wider window of β-AR activity cannot yet be excluded (0–3 h post-reactivation). The administration of propranolol 1 h before reactivation may have blocked early β-AR activity, whereas propranolol administration 1 h post-reactivation may have blocked late β-AR activity. However, in support of the hypothetical late β-AR involvement, offline activity of norepinephrine-containing locus coeruleus neurons and norepinephrine release ~2 h after learning was associated with memory consolidation of odor discrimination in rats[53]. A similar time-window of ~2 h for locus coeruleus activity was observed after reactivation of remote memory, suggesting a late noradrenergic involvement in memory reconsolidation as well[53]. Nevertheless, the alleged time-window of 2–3 h post-reactivation is in sharp contrast with other findings on memory consolidation, where transient β-AR activity before or during learning, but not after learning, sets in motion cascading molecular events for the formation (AMPARs) and subsequent consolidation of long-term memory (PKA and ERK)[20, 21, 54]. It is worth noting here that in addition to the common neurobiological mechanisms of consolidation and reconsolidation[38], distinct molecular processes and brain regions in these memory processes are also reported[55, 56]. On basis of the current observations we suggest for fear memory reconsolidation in humans that transient but late β-AR activity exerts metaplastic effects for a former destabilized memory. Interestingly, delayed hippocampal ERK activity also around 3 h is involved in memory reconsolidation of inhibitory avoidance learning in mice[57]. Irrespective of this striking time-overlap between delayed β-AR and ERK activity, further research is needed to demarcate the exact time-window of β-AR activity in the reconsolidation of memory. Another issue to be addressed in future research is whether a similar β-AR window can be observed across species and learning tasks.

In line with the reconsolidation hypothesis, we show intact fear memory expression in humans following the reactivation and amnesic drug intake, with subsequent amnesia. We demonstrate that the mnemonic output of associative fear learning may remain functional for at least 12 h after the reconsolidation intervention, as long as the retention test took place during the same day of the intervention. The most notable finding of the current study was that the absence of fear expression became only apparent after a night of sleep (either 12 or 24 h later). These findings corroborate the hypothesis that sleep is not only involved in the processes of memory consolidation, but also in reconsolidation[58]. An alternative explanation for the current observation is also worth considering, given that the active phase of propranolol (i.e., half-life of 5 h) fell mainly in the first hours of the night for the sleep group, whereas propranolol was active in the wake state

for the no-sleep group. Hence, propranolol may have exerted its effect through mimicking or even boosting noradrenergic silence in the locus coeruleus that has been associated with targeted depotentiation and subsequent forgetting during sleep[59]. Yet, this alternative hypothesis of noradrenergic silence in the locus coeruleus struggles to explain both the frequently observed post-reactivation amnesia for learned fear in our laboratory[8–17] and the current observation of post-reactivation amnesia in the no-sleep group at the second retention test 24 h later. Although the no-sleep group initially showed intact memory expression when tested 12 h later at the same day of the intervention, they actually showed amnesia for learned fear when they were tested again, 24 h after drug intake. Given that propranolol was no longer active during the sleep state for the no-sleep group, the alternative hypothesis cannot explain the post-reactivation amnesia at the second retention test. In addition, in our previous studies (including exp_1 and exp_2) post-reactivation amnesia has been convincingly observed 24 h later, while propranolol had mainly been active during the day[8–17].

Irrespective of noradrenergic activity, the selection of keeping and deleting memories is thought to be an active and well-regulated process, which is presumably unique to sleep[59, 60]. Although periods of wakefulness are associated with a net increase in synaptic strength, periods of sleep are associated with a net decrease, thereby preserving an overall balance of synaptic strength[61–64]. At the molecular level, it has been shown that GluR1-containing AMPAR levels are high during waking and low during sleep across the entire cortex[61]. Likewise, the slope of cortical evoked responses—an electrophysiological marker of synaptic efficacy—increased after wakefulness and decreased after sleep[61]. These relative changes in molecular and electrophysiological indicators of synaptic strength are largely independent of the time of day[61]. Synaptic renormalization seems to affect a majority of synapses, but it should also be selective to allow for both stability and plasticity[62, 64]. If the process of memory reconsolidation has been disrupted by either a pharmacological or behavioral intervention, a targeted synaptic downscaling of the fear memory engram may basically take place when the organism is disconnected from the environment. As a consequence, post-reactivation amnesia should only become apparent after sleep. For many forms of memory, brief naps of 40–90 min are sufficient to promote consolidation and reconsolidation processes, suggesting that not circadian rhythm—but sleep itself—is imperative for memory and forgetting[50–52, 65–67]. An interesting question for future research is whether a brief nap instead of a proper night of sleep would also suffice to trigger pharmacologically induced post-reactivation amnesia for learned fear. Though we can merely speculate at this stage, our findings suggest that reconsolidation is not simply time-dependent, as has frequently been claimed in the animal literature, but that sleep may be a final and necessary link to complete the process of memory reconsolidation.

## Methods

**Blocking central β-ARs disrupts fear memory reconsolidation**. Participants (n = 30) underwent a differential fear-conditioning procedure that included different phases across three days (see Fig. 1a for design and Supplementary Table 1 for participant characteristics). Participants received either experimental-credits or a small amount of €50 for their participation in the study. Informed consent was obtained from all participants and the ethical board of the University of Amsterdam approved the study.

Conditioned fear responding was measured as potentiation of the eye-blink startle reflex to a loud noise by electromyography of the right orbicularis oculi muscle. Startle potentiation is regarded a specific as well as reliable index of fear that is modulated by and directly connected with the amygdala[68]. During acquisition a CS1 was repeatedly paired with an US whereas a CS2 was not. Fear-relevant stimuli were utilized (i.e., pictures of spiders—International Affective Picture System (IAPS) numbers[69] 1200 and 1201) as they result in stronger

conditioning and are quite resistant to extinction learning[70]. Moreover, given that most anxiety disorders are associated with this type of stimulus categories, we are specifically interested in targeting stronger fear memory.

CSs were presented in the middle of a computer screen. We counterbalanced the assignment of the slides as CS1 and CS2 across participants. All stimuli were presented for 8 s. Startle probes (i.e., loud noises of 40 ms and 104 dB) were presented 7 s after CS onset through headphones and were followed by the US 500 ms later for the CS1. We used an electric stimulus with duration of 2 ms as an US, which was delivered to the wrist of the left hand via a pair of Ag electrodes of 20 by 25 mm with fixed inter-electrodes mid-distances of 45 mm. A conductive-gel was applied between the electrodes and the skin. Delivery of the US was controlled by a Digitimer DS7A constant current stimulator.

We individually set the US intensity at a level described by the participants as "uncomfortable but not painful". The US remained set to this intensity throughout the experiment. Starting at an intensity of 1 mA the level of the 2-ms electric stimulus was gradually increased. After US selection, two 7-mm sintered Ag-AgCl electrodes filled with electrolyte gel were positioned approximately 1 cm under the pupil and 1 cm below the lateral canthus for the EMG recordings and a ground reference was placed on the forehead[71]. Next, we informed the participants regarding the CSs: they were told that an electric stimulus would follow one of the slides in most cases and that the other slide would never be followed by the electric stimulus. Moreover they were instructed to learn to predict whether the electric stimulus would occur or not on basis of the slides. During the presentation of each slide, the participants had to rate their expectancy of the electric stimulus by shifting a cursor on a continuous 11-point scale labeled from "certainly no electric stimulus" through "uncertain" to "certainly an electric stimulus" and push the left mouse button within 5 s following stimulus onset.

All three sessions began with a habituation phase consisting of 10 startle probes to reduce initial startle reactivity. During fear acquisition the CS1 was presented 5 times on an 80% reinforcement scheme with the first trial unreinforced. Furthermore, 5 unreinforced CS2 trials and 5 baseline startle probes (i.e., noise alone, NA trials) were presented alone during the inter-trial intervals, which varied between 15 and 20 and 25 s with a mean of 20 s. Order of trial type was randomized within blocks of 3 trials (i.e., CS1, CS2, and NA) such that no more than two trials of the same type occurred in succession.

After fear acquisition a break of 24 h was inserted in order to substantiate consolidation of the fear memory. Furthermore, participants were instructed to refrain from caffeine and alcohol during the 12 h and to avoid food and drinks other than water during the 2 h prior to the next session. Given the peak-plasma concentrations of the drugs[42, 43], propranolol (40 mg—$n = 15$) was provided 1 h and nadolol (40 mg—$n = 15$) 3 h before MR. Both drugs were administered double-blind and placebo-controlled. Before reactivation, we told the participants that the same two slides would be presented and we asked them to remember what they had learned the previous day. Instructions about the US-expectancy ratings were comparable to acquisition. During MR a single CS1-R without the US was presented, which was followed by a NA startle probe. For ensuring that the two drugs exerted a similar physiological effect both blood pressure and heart rate levels were obtained before the first pill administration and upon completion of the session.

Considering the elimination half-life of the drugs[42, 43], memory retention was tested 1 week following reactivation allowing the drugs to wash out. The participants were told that the same two pictures provided during acquisition would be presented and they were asked to rate the expectancy of the US during the presentation of all slides. During extinction learning participants were exposed to the CS1, CS2, and the NA for 10 times without the US. For maximizing the likelihood of fear memory expression, we administered three unsignaled reminder shocks following extinction learning. The timing between the test trials and the reinstating USs was 19 s. Subsequent to the unsignaled USs, we again presented the participants with five CS1, CS2, and NA trials. The timing between the unsignaled shocks and reinstatement testing was 18 s.

**β-ARs are critical within a specific time-window**. Participants ($n = 20$) were subjected to a differential fear-conditioning paradigm that allowed for selectively reactivating one of two distinct fear memories and included several phases across three subsequent days (see Fig. 2a for design and Supplementary Table 1 for participant characteristics).

Conditioned fear responding was again measured by startle potentiation: see experiment 1 for details. Whereas two fear-relevant stimuli of different stimulus categories served as the CS1 and CS2 (i.e., pictures of a spider and gun—IAPS numbers[69] 1200 and 6210), a fear-irrelevant stimulus served as CS3 (i.e., a mug— IAPS number 7009). Our CS2 fear-relevant stimulus served as control for the CS1 fear-relevant stimulus. Yet, as fear-relevant stimuli are known to have an inherent prepotency to elicit fear responses[70], we also employed a fear-irrelevant control cue to test whether disrupting reconsolidation could neutralize the acquired fear responding. We counterbalanced the assignment of the slides as CS1 and CS2 across participants. All stimuli were presented for 8 s. Startle probes were presented 7 s after CS onset and were followed by the US 500 ms later for the CS1 and CS2: see experiment 1 for details on the conditioning procedure.

For obtaining a baseline measurement in physiological responding, the US electrodes were kept detached during the first part of fear acquisition. After attachment of the Ag-AgCl electrodes we informed the participants about the CSs:

they were told that pictures of a spider, gun and mug would be shown on the computer screen and that no electric stimuli could be delivered yet. During this first part of fear acquisition both the three stimuli and the NA were presented just once. Before the actual fear learning the US electrodes were attached and we determined the intensity of the US: see experiment 1 for details. After US selection we instructed the participants to look carefully at the slides. They were told that an an electric stimulus would follow two of the slides in all cases and that the third slide would never be followed by the electric stimulus. Moreover they had to learn to predict the occurrence of the electric stimulus on basis of the slides. Furthermore, the participants were asked to rate the expectancy of the US during the presentation of each slide: see also experiment 1 for details. During this second part of fear acquisition the CS1, CS2, and CS3 were presented three times. Furthermore, three baseline startle probes were presented alone during the inter-trial-intervals.

After fear acquisition a break of about 24 h was inserted in order to support consolidation of the fear memory. Moreover, participants were instructed to refrain from caffeine and alcohol during the 12 h and to avoid any food and drinks other than water during the 2 h before MR. After the attachment of the electrodes, the participants were told that the same three slides would be presented on the computer screen. In addition, we asked them to remember what they had learned the previous day. Further instructions about the US-expectancy ratings were comparable to acquisition. During MR a single unreinforced CS1-R was presented followed by a NA startle probe. Participants in the pill_1h group ($n = 10$) and pill_2h group ($n = 10$) received an oral dose of 40 mg of propranolol exactly 1 or 2 h after MR respectively, in a single-blind fashion (i.e., the participants were blind to medication assignment: propranolol HCl vs. pill placebo even though all participants got the active drug).

Memory retention was tested about 24 hr after reactivation. Instructions regarding the CSs merely revealed that again the same three pictures would be presented. Participants were also instructed to report the expectancy of the US. During extinction learning the participants were exposed to the CS1, CS2, and CS3 for six times without the US. Six startle probes were also presented alone. Three unsignaled "reminder shocks" were administered following extinction learning, and the timing between the test trials and the reinstating US was 19 s. Subsequent to the unsignaled shocks participants were again presented with three CS1, CS2, CS3, and NA trials. The timing between the unsignaled shocks and reinstatement testing was 18 seconds.

**Post-reactivation amnesia requires sleep**. Participants ($n = 20$) were subjected to a similar fear-conditioning procedure as described in experiment 2 (see also Fig. 3a for design and Supplementary table 1 for participant characteristics). However, for the participants in the no_sleep group ($n = 10$) and sleep group ($n = 10$) the memory was reactivated in the morning or evening time respectively. All of the participants received single-blind an oral dose of 40 mg of propranolol (i.e., the participants were blind to medication assignment) right after reactivation of the memory. A first memory retention test was inserted exactly 12 h following reactivation: in the evening on day 2 for the participants in the no_sleep group and in the morning on day 3 for the participants in the sleep group. Instructions regarding the CSs merely revealed that again the same three pictures would be presented. Participants were instructed to report the expectancy of the US. During this first memory retention test the CS1, CS2, CS3, and NA were presented just once without the US. Precisely 24 h later memory retention was again tested: in the evening on day 3 for the participants in the no_sleep group and in the morning on day 4 for the participants in the sleep group. See experiment 2 for the procedure of this second memory retention test.

**EMG recordings and statistical analysis**. We measured the eye-blink EMG activity by using a bundled pair of electrodes wires, which were connected to a front-end amplifier with input resistances of 10 MΩ and a bandwidth of DC-1500 Hz. For reducing unwanted interferences, we set a notch filter at 50 Hz. Integration was handled by a true-RMS converter with a time constant of 25 ms. Integrated EMG signals were sampled at 1000 Hz. Baseline-to-peak amplitudes were identified over the period of 50—150 ms following probe onset. Startle responses as well as US-expectancy ratings were analyzed by means of a mixed analysis of variances for repeated measures with group [i.e., exp_1: propranolol vs. nadolol, exp_2: pill_1h vs. pill_2h; exp_3: no_sleep vs. sleep] as between-subjects factor and stimulus [i.e., exp_1: CS1 vs. CS2; exp_2 and exp_3: CS1 vs. CS3 and CS2 vs. CS3] and trial [i.e., stimulus presentation] as within-subjects factors. We compared the differential responding over the testing phases respectively [i.e., first trial vs. last trial]. Planned comparisons between the CS1, CS2, and CS3 stimuli were performed separately. Missing startle responses caused by recording artifacts or trials with excessive baseline activity (i.e., 0.5%) were excluded from analyses. Given that the maximum of invalid responses within-subjects was only 7%, we did not exclude any participant from the dataset. Significance was set at $P < 0.05$.

**Data availability**. The data that support the findings of this study are available from the corresponding author upon reasonable request

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

## Acknowledgements

We thank Bert Molenkamp for technical assistance and Dieuwke Sevenster for checking the data processing and statistical analyses. During data collection and data analysis at the University of Amsterdam, M.K. and M.S were supported by the Netherlands Organization for Scientific Research.

## Author contributions

M.K. and M.S. designed the studies. M.S. collected and analyzed the data. M.K. and M.S. wrote the paper.

## Additional information

**Competing interests:** The authors declare no competing interests.

