## [Peer Review File(PDF 437 kb) · Nature Communications]

Reviewers' comments:

Reviewer #1 (Remarks to the Author):

Kindt and Soeter present an interesting set of experiments revealing three prerequisites for pharmacological interference with reconsolidation of fear-conditioned memory. Each experiment employs a systemic administration of a noradrenergic antagonist and employs a peripheral readout measurement (startle responses).

The results delineate three central boundary conditions of reconsolidation processes in humans to the current literature.

The authors are complimented on their elegant design, which was inspired by theories and rodent work on synaptic processes during memory consolidation. However, the authors need to be careful to avoid overinterpretation of the results (see comment 4) in terms of reverse inference (e.g. synaptic plasticity involves noradrenaline, but noradrenaline does not only work on synaptic plasticity). I am aware that systemic administration of drugs is "the best shot" to infer central processes in humans, hence the authors might reconsider framing the experiments.

Another major concern is the small sample size, which can be easily addressed by providing a power calculation.

1) In case of non significant findings, the authors provide no p-values, but F-values. In the cases of low F-values this strategy is understandable and even more informative (e.g. $F < 1$). But for higher F-values (e.g. $F(1,128) < 3.17$ on page 7 or $F(1,9) = 4.36$ on page 10), it is hard to infer p-values ($p = 0.077$ and $p = 0.065$, respectively) from these F-values. Since p-values are still the standard in reporting statistical results, I would advocate for reporting all p-values below 0.1 in addition to the F-values. Actually, p-values below 0.1 in such small sample sizes are sometimes considered as "statistical trends".

2) While the conclusion is parsimonious that sleep might be causal for the reconsolidation of the memory trace, the authors need to be more careful with this interpretation. In particular, it might well be that differences in endogenous regulation of neurotransmitters like cortisol, melatonin, etc through the circadian rhythm could have interfered with the reconsolidation. Hence, since sleep is one of a bunch of differences between groups, the authors should be careful with pinpointing a monocausal influence. Related to this, I did not understand the conclusion of this study, that "post retrieval amnesia was independent of time" (p.13 line 277).

3) The authors make several assumptions on the synaptic level that simply cannot be tested in their experiments. I suggest that the authors tone down some of their assumptions, especially in the results section. For example, Page 5 line 98: "If the administration of propranolol-HLC disrupts the production of PRPs independent from affecting the re-learning tag ...". The authors do not test if propranolol affects synaptic proteins or memory tags, but a prerequisite: crossing the blood-brain barrier. The interpretation of the results (e.g. page 8 line 155) is well in line with experimental design and do not mention synaptic tags

anymore.

Another example can be found on page 11, l.225. The description of the third experiment starts with : " To establish whether the beta-AR blocker propranolol acts specifically on the molecular mechanism mediating reconsolidation...". This has not been tested in experiment three.

An example for a clear overstatement is found in the discussion (page 15, l.332): "Our findings suggest that that blocking beta-AR activity subsequent to memory destabilization prevents the synthesis of proteins". Protein synthesis has not been examined in this study. Rephrasing these examples and other parts in the manuscript would reframe the experiments into the right setting, i.e. systemic pharmacological challenge during a behavioural manipulation.

4) The authors do not report any exclusion of datasets in their analyses. Given that startle signals are not always perfect, I wonder if any participants were excluded or pre-selected?

5) The results reported for experiment 2 and 3 (employing two CSs paired with an US) are missing the comparison between the reactivated CS against the non-reactivated (e.g. CS1 vs CS2). Instead comparisons between the CS associated with the US and the control stimulus are employed. However, these comparisons do not reveal the effect of reactivation exerted on a CS that has been previously associated with the US. Hence, comparisons between these stimuli need to be included.

6) On page 10 line 201, the authors state that planned comparisons (CS1 vs CS3) revealed reduced memory retention in the 1hour pill group and intact differentiation in the 2hour pill group. Is the difference (or interaction) between groups significant, as well? Or is the interaction with the factor group only significant during extinction? The same is true for the planned comparisons (CS1 vs CS3) between the sleep and no sleep group on page 12 line 251. Here again, the crucial group comparison is not reported.

7) Propranolol is a classical example of a non-selective Antagonist at the beta Adrenoreceptor. However, Propranolol has antagonistic properties at 5-HT1 receptor (5-HT1B, 5-HT1A , 5-HT1c; Nishio H, Nagakura Y & Segawa T 1989; Arch Int Pharmacodyn 302:96-106), as well. Agren et al found an impact on 5HT on reconsolidation as well (Translational Psychiatry 2012). Hence, it might well be that 5HT contributes besides NA to reconsolidation in humans. In fact Nadolol has no affinity to 5HT receptors, which might have contributed to the current results.

Minor Comments:

- The subjective evaluation of US intensities is around 3. Please provide the range of that scale.

-How many participants in each experiment were aware of the stimulus contingencies?

- Please report the gender of the participants included in each sample and provide some evidence (e.g. testing in one of the experiments) that gender did not influence reconsolidation.

-Please provide information about the intake of food in each group, especially in the comparison between different time-points of drug administration. Food might have influenced the pharmacokinetic profiles.

-Please consider exchanging "biochemical signature" in the manuscript (e.g. page 8 line 173) with "pharmacokinetic profile/signature". While this might sound picky, it is just a meant to be a constructive help to use standardized terms.

Reviewer #2 (Remarks to the Author):

In 3 independent experiments the authors show in human participants that the expression of conditioned fear memories (measured via startle response potentiation) can be abolished if propranolol (a β -adrenergic antagonist) is administered during a specific time window following memory reactivation. Moreover, the reduction of fear responses on the long-term was found to be dependent on intervening sleep. This is a nicely designed and well-controlled study that tackles a timely and important issue. The methods are sound and the manuscript is well-written. However, I have a few concerns that should be addressed.

1. My main concern is with regard to the authors' conclusion of a specific time window for β -AR activity of 2-3 hours post-reactivation (Fig. 3). In my understanding of the data, it cannot be excluded that β -AR activity during the first 1-2 hours plays a role for memory reconsolidation as well. What the data clearly show is that after 3 hours β -AR activity is (probably) no longer required. (Although even this conclusion is shaky because later intervals were not tested, e.g. it is possible that there is a second time window, e.g. around 6 hours post-reactivation, with a β -AR dependency.) But more importantly, based on Fig. 3 it cannot be excluded that propranolol acted on reconsolidation processes during the first 2 hours after reactivation. The drug administered 1 h before and right after reactivation could also have exerted an effect sooner than the 2-3 h time window. In order to show that in this phase β -AR activity is not required for reconsolidation, earlier drug administration would have to be tested, e.g. 2 vs. 5 hours before reactivation. This issue should be appropriately discussed and the conclusions toned down.

2. The observed sleep effect is particularly novel and interesting and I wonder about the possible underlying mechanisms of this effect. The authors discuss synaptic downscaling as one possibility, which seems to be reasonable. An alternative mechanism that came to my mind is the formation and preferential consolidation of a new memory trace. During the reactivation session, participants may acquire a new safety memory trace, i.e. learning that the CS1 is no longer paired with the US. Sleep may then foster the preferential consolidation of this new safety memory trace that interferes with the fear memory trace.

3. The sleep and no sleep groups of Exp. 3 differ in several regards. First, considering that propranolol was administered after reactivation, the active phase of the drug fell mainly in

the sleep state for the sleep group and in the wake state for the no sleep group. Do the authors know of any evidence showing that propranolol may act differently during sleep and wakefulness? A number of hormones, neurotransmitters and plasticity-related processes differ between sleep and wakefulness and, thus, the effects of propranolol may also differ. Second, when did acquisition/extinction take place for the sleep and no sleep groups? If it occurred at the same time of day for both groups, this would mean that there were differences in the retention interval between acquisition and reactivation as well as between test and extinction for the two groups. Alternatively, if acquisition/extinction took place also in the evening and morning, respectively, can the authors exclude any potential circadian effects?

4. A few methodological details should be clarified:

- o In Exp. 1, in the instruction for participants it says that "an electric shock would follow one of the slides in most cases". Was the CS1 actually followed by an electric shock in all cases? Why were participants instructed otherwise? And why was the instruction different in Exp. 2, where it says "in all cases"?

- o Line 376: what does randomized within blocks (i.e., CS1 – CS2 – NA) mean? Were stimuli of each category always presented consecutively in one block, i.e. all CS1 one after the other etc.?

- o Line 385/396: were CSs presented without US here?

- o It is said that the drug was applied in a single blind fashion? What does that mean? Who was blind, the participants or the experimenter?

- o Missing startle responses (0.5%) were excluded. How did the authors define missing startle responses?

Minor comments:

1. Parts of the introduction, especially with regard to the details of the molecular pathways, do not seem to be essential for the rationale of the experiments. Perhaps these parts could be moved to the discussion section and the introduction could be shortened a bit.

2. It would be helpful to spell out β -AR in the title for non-experts. Also, the abbreviation PE in extended Figures 1, 2 and 3 should be defined.

3. Line 136: The abbreviation CS1-R is not defined here. Could it simply be termed CS1 here?

4. Line 190: after reactivation instead of retrieval?

5. Typos: line 305 "This in", line 322 "took play", line 364 "a ground references", line 709 "extinction trial group"

Reviewer 1:

Kindt and Soeter present an interesting set of experiments revealing three prerequisites for pharmacological interference with reconsolidation of fear-conditioned memory. Each experiment employs a systemic administration of a noradrenergic antagonist and employs a peripheral readout measurement (startle responses). The results delineate three central boundary conditions of reconsolidation processes in humans to the current literature. The authors are complimented on their elegant design, which was inspired by theories and rodent work on synaptic processes during memory consolidation. However, the authors need to be careful to avoid overinterpretation of the results (see comment 4) in terms of reverse inference (e.g. synaptic plasticity involves noradrenaline, but noradrenaline does not only work on synaptic plasticity). I am aware that systemic administration of drugs is “the best shot” to infer central processes in humans, hence the authors might reconsider framing the experiments. Another major concern is the small sample size, which can be easily addressed by providing a power calculation.

We agree with the reviewer that we should avoid reverse inference (i.e., inferring biological processes from behavioural data). We have now changed our text accordingly throughout the manuscript (see response to comment 4). Another major concern raised by this reviewer is the small sample size, which could indeed be easily addressed by a power calculation. Hence, following the suggestion of the reviewer, we have now added the power analysis in a footnote at the bottom of page 9. This yielded a sample size of 20 participants for a large effect size of $f = 0.35$ with an alpha level of 0.05 and a power level of 0.95. Given the large effect size, we believe that the current sample sizes are sufficient.

1 – In case of non significant findings, the authors provide no p-values, but F-values. In the cases of low F-values this strategy is understandable and even more informative (e.g. $F < 1$). But for higher F-values (e.g. $F(1,128) < 3.17$ on page 7 or $F(1,9) = 4.36$ on page 10), it is hard to infer p-values ($p = 0.077$ and $p = 0.065$, respectively) from these F-values. Since p-values are still the standard in reporting statistical results, I would advocate for reporting all p-values below 0.1 in addition to the F-values. Actually, p-values below 0.1 in such small sample sizes are sometimes considered as “statistical trends”.

We have now reported all p -values below 0.1 in addition to the F -values and commented on the statistical trends as well – see lines 138-139 and line 213-214.

2 – While the conclusion is parsimonious that sleep might be causal for the reconsolidation of the memory trace, the authors need to be more careful with this interpretation. In particular, it might well be that differences in endogenous regulation of neurotransmitters like cortisol, melatonin, etc through the circadian rhythm could have interfered with the reconsolidation. Hence, since sleep is one of a bunch of differences between groups, the authors should be careful with pinpointing a monocausal influence. Related to this, I did not understand the conclusion of this study, that “post retrieval amnesia was independent of time” (p.13 line 277).

We agree with the reviewer that we should be careful with our conclusions on the role of sleep and that post-reactivation amnesia is independent of time. We have now deleted this sentence from the manuscript and toned down the discussion (see lines 304-306 and 314-315). We have elaborated more extensively on the role of sleep in memory (re)consolidation in the general discussion (pages 16-17, lines 370-387, lines 396-400). Although the current study did not test for the effect of circadian rhythm, we refer to literature suggesting that it is not very likely that the relation of sleep and memory is explained by circadian rhythm (lines 312-315).

3 – The authors make several assumptions on the synaptic level that simply cannot be tested in their experiments. I suggest that the authors tone down some of their assumptions, especially in the results section. For example, Page 5 line 98: “If the administration of propranolol-HLC disrupts the production of PRPs independent from affecting the re-learning tag ...”. The authors do not test if propranolol affects synaptic proteins or memory tags, but a prerequisite: crossing the blood-brain barrier. The interpretation of the results (e.g. page 8 line 155) is well in line with experimental design and do not mention synaptic tags anymore.

Another example can be found on page 11, l.225. The description of the third experiment starts with : “ To establish whether the beta-AR blocker propranolol acts specifically on the molecular mechanism mediating reconsolidation...”. This has not been tested in experiment three.

An example for a clear overstatement is found in the discussion (page 15, l.332): “Our findings suggest that that blocking beta-AR activity subsequent to memory destabilization prevents the synthesis of proteins”. Protein synthesis has not been examined in this study. Rephrasing these examples and other parts in the manuscript would reframe the experiments into the right setting, i.e. systemic pharmacological challenge during a behavioural manipulation.

We have now changed the text on page 5-6, lines 100 to 104. Furthermore, we have replaced the word ‘predictions’ on line 76 in the introduction by ‘hypotheses’. We also changed the text on page 11, lines 245 to 248.

Although we agree with the reviewer that we do not directly test whether propranolol affects the synaptic tag or protein synthesis, we actually formulated our hypotheses and design on basis of these processes. By simply ignoring these processes, the current designs and hypotheses would be hard to fathom. We agree though that we can only infer the critical conditions to target and observe memory reconsolidation from the underlying neurobiological processes. Therefore, we have now removed these inferences from the predictions and result descriptions in our manuscript.

4 – The authors do no report any exclusion of datasets in their analyses. Given that startle signals are not always perfect, I wonder if any participants were excluded or pre-selected?

Missing startle responses caused by recording artifacts or trials with excessive baseline activity (i.e., 0.5%) were excluded from the analyses. We explain this in lines 542-543.

5 – The results reported for experiment 2 and 3 (employing two CSs paired with an US) are

missing the comparison between the reactivated CS against the non-reactivated (e.g. CS1 vs CS2). Instead comparisons between the CS associated with the US and the control stimulus are employed. However, these comparisons do not reveal the effect of reactivation exerted on a CS that has been previously associated with the US. Hence, comparisons between these stimuli need to be included.

Following the suggestion of the reviewer, we have now included the comparisons between the reactivated CS1 and non-reactivated CS2 stimuli for both experiment_2 and experiment_3 – see pages 10-13 .

6 – On page 10 line 201, the authors state that planned comparisons (CS1 vs CS3) revealed reduced memory retention in the 1hour pill group and intact differentiation in the 2hour pill group. Is the difference (or interaction) between groups significant, as well? Or is the interaction with the factor group only significant during extinction? The same is true for the planned comparisons (CS1 vs CS3) between the sleep and no sleep group on page 12 line 251. Here again, the crucial group comparison is not reported.

Indeed, the interactions between groups are significant as well, which is reported in lines 210-214 and 272-275. For reasons of clarification, it is now stated that “Planned comparisons *in fact* show. ed..” in line 210-211 and “Planned comparisons *indeed* showed..” in line 272.

7 – Propranolol is a classical example of a non-selective Antagonist at the beta Adrenoreceptor. However, Propranolol has antagonistic properties at 5-HT1 receptor (5-HT1B, 5-HT1A , 5-HT1c; Nishio H, Nagakura Y & Segawa T 1989; Arch Int Pharmacodyn 302:96-106), as well. Agren et al found an impact on 5HT on reconsolidation as well (Translational Psychiatry 2012). Hence, it might well be that 5HT contributes besides NA to reconsolidation in humans. In fact Nadolol has no affinity to 5HT receptors, which might have contributed to the current results.

Although we cannot measure whether the effects of propranolol in our human participants can be explained by blocking the β -ARs, there is ample and compelling evidence from the animal literature showing that the effect of propranolol in fear memory consolidation actually works via β -ARs (e.g., Johansen, et al. 2014, Hebbian and neuromodulatory mechanisms interact to trigger associative memory formation. *PNAS* 111, 5584-5592; Tenorio et al. 2010, ‘Silent’priming of translation-dependent LTP by β -adrenergic receptors involves phosphorylation and recruitment of AMPA receptors. *Learn. Mem.* 17, 627-638). The study by Agren suggests that dopaminergic and serotonergic genes influence human fear memory reconsolidation, but these findings have neither been replicated, nor are they supported by animal research. More importantly, Agren et al did not manipulate memory reconsolidation by the administration of propranolol, but they tested the effect of the retrieval-extinction procedure, which is very different from pharmacologically inducing post-reactivation amnesia (see also Beckers & Kindt, 2017, *Ann. Rev. Clin. Psychol.* 13, 1).

Minor Comments:

1. The subjective evaluation of US intensities is around 3. Please provide the range of that scale. US evaluation scores ranged from 0 to 5, where higher scores indicate more aversive, (see line 759).

2. How many participants in each experiment were aware of the stimulus contingencies? All participants were aware of the CS-US contingencies immediately after they underwent the fear conditioning procedure. We now clarify this in the legends of the Extended Data Figs. 1-3 on pages 34-36.

3. Please report the gender of the participants included in each sample and provide some evidence (e.g. testing in one of the experiments) that gender did not influence reconsolidation. We now report the gender of the participants included in each of the experiments (line 108, line 193 and line 256). However, the present sample sizes are too small to properly test for any gender effects. But note that in our previous studies we never detected differences between the gender groups. We have now clarified this in a footnote at the bottom on page 6.

4. Please provide information about the intake of food in each group, especially in the comparison between different time-points of drug administration. Food might have influenced the pharmacokinetic profiles. After acquisition, participants were instructed to refrain from caffeine and alcohol during the 12 h and to avoid food and drinks other than water during the 2 h prior to memory reactivation. We now clarify this on page 19 and page 21.

5. Please consider exchanging “biochemical signature” in the manuscript (e.g. page 8 line 173) with “pharmacokinetic profile/signature”. While this might sound picky, it is just a meant to be a constructive help to use standardized terms. We have now exchanged “biochemical signature” with “pharmacokinetic signature” throughout the manuscript (line 17, line 177, line 181, line 235, line 332, line 738).

Reviewer 2:

In 3 independent experiments the authors show in human participants that the expression of conditioned fear memories (measured via startle response potentiation) can be abolished if propranolol (a β -adrenergic antagonist) is administered during a specific time window following memory reactivation. Moreover, the reduction of fear responses on the long-term was found to be dependent on intervening sleep. This is a nicely designed and well-controlled study that tackles a timely and important issue. The methods are sound and the manuscript is well-written. However, I have a few concerns that should be addressed.

1 – My main concern is with regard to the authors’ conclusion of a specific time window for β -AR activity of 2-3 hours post-reactivation (Fig. 3). In my understanding of the data, it cannot be excluded that β -AR activity during the first 1-2 hours plays a role for memory reconsolidation as well. What the data clearly show is that after 3 hours β -AR activity is (probably) no longer required. (Although even this conclusion is shaky because later intervals

were not tested, e.g. it is possible that there is a second time window, e.g. around 6 hours post-reactivation, with a β -AR dependency.) But more importantly, based on Fig. 3 it cannot be excluded that propranolol acted on reconsolidation processes during the first 2 hours after reactivation. The drug administered 1 h before and right after reactivation could also have exerted an effect sooner than the 2-3 h time window. In order to show that in this phase β -AR activity is not required for reconsolidation, earlier drug administration would have to be tested, e.g. 2 vs. 5 hours before reactivation. This issue should be appropriately discussed and the conclusions toned down.

We inferred the specific time window of β -AR activity by combining the different timings of drug administration, the pharmacokinetics of propranolol and the presence/absence of changing the expression of fear memory. This has now been explained in more detail in the discussion on page 15, lines 331-349.

2 – The observed sleep effect is particularly novel and interesting and I wonder about the possible underlying mechanisms of this effect. The authors discuss synaptic downscaling as one possibility, which seems to be reasonable. An alternative mechanism that came to my mind is the formation and preferential consolidation of a new memory trace. During the reactivation session, participants may acquire a new safety memory trace, i.e. learning that the CS1 is no longer paired with the US. Sleep may then foster the preferential consolidation of this new safety memory trace that interferes with the fear memory trace.

It is highly unlikely that 1 unreinforced trial will trigger extinction learning thereby forming an inhibitory or safety memory. Actually the data in the non-effective conditions show that 1 unreinforced trial did not trigger extinction learning: The startle fear response from acquisition to test after one unreinforced trial did not decline at all (i.e., nadolol group in Fig. 1, pill_2h group in Figure 2, or the first retention test 12 h later without sleep in between in Fig. 4). In our previous work we have also demonstrated that a reinforced trial CS+ followed by propranolol neutralized the fear memory the following day, as long as the memory reactivation involves a prediction error (see Sevenster, Beckers & Kindt, 2013, *Science*, 339, 830-833). These observations can be considered as a falsification for the alternative hypothesis of inhibitory learning as opposed to weakening the excitatory fear memory.

3 – The sleep and no sleep groups of Exp. 3 differ in several regards. First, considering that propranolol was administered after reactivation, the active phase of the drug fell mainly in the sleep state for the sleep group and in the wake state for the no sleep group. Do the authors know of any evidence showing that propranolol may act differently during sleep and wakefulness? A number of hormones, neurotransmitters and plasticity-related processes differ between sleep and wakefulness and, thus, the effects of propranolol may also differ. Second, when did acquisition/extinction take place for the sleep and no sleep groups? If it occurred at the same time of day for both groups, this would mean that there were differences in the retention interval between acquisition and reactivation as well as between test and extinction for the two groups. Alternatively, if acquisition/extinction took place also in the evening and morning, respectively, can the authors exclude any potential circadian effects?

Although the sleep and non-sleep conditions differ indeed in several ways, there are strong arguments why these differences cannot explain the current observations. Even though the fear-reducing effects were initially not observed in the group who had the memory reactivation in the morning and a retention test 12 h later at the same day, this group also showed amnesia for learned fear when they were again tested 24 h later after a night of sleep. In view of the bioavailability of propranolol during the first hours of the night in the sleep-group, the drug was no longer available in the other no-sleep group, while they showed a similar fear reduction when tested 24 h later. This has now been extensively explained in the discussion section, see pages 16-17, lines 370-387, lines 396-400. Although the current study did not test for the effect of circadian rhythm, we refer to literature suggesting that it is not very likely that the relation of sleep and memory is explained by circadian rhythm (lines 312-315).

A few methodological details should be clarified:

- **In Exp. 1, in the instruction for participants it says that “an electric shock would follow one of the slides in most cases”. Was the CS1 actually followed by an electric shock in all cases? Why were participants instructed otherwise? And why was the instruction different in Exp. 2, where it says “in all cases”?** In exp_1, the CS1 was in fact followed by the US on an 80% reinforcement scheme (i.e., non-asymptotic learning), which we have now clarified in lines 438-439. But in exp_2 and exp_3, the CS1 and CS2 were followed by the US in all cases (i.e., on a 100% reinforcement scheme or asymptotic learning).
- **Line 376: what does randomized within blocks (i.e., CS1 – CS2 – NA) mean? Were stimuli of each category always presented consecutively in one block, i.e. all CS1 one after the other etc.?** We now clarify that order of trial type was randomized within blocks of 3 trials (i.e., CS1, CS2, and NA) such that no more than two trials of the same type occurred in succession – see lines 441-443.
- **Line 385/396: were CSs presented without US here?** In lines 453-454 and lines 502-503 it is now clarified that during MR as well as during extinction learning the CSs were presented without the US.
- **It is said that the drug was applied in a single blind fashion? What does that mean? Who was blind, the participants or the experimenter?** We now clarify in lines 450-451 (double blind), lines 505-506 (single blind), lines 519-520 (single blind) that the participants were blind to medication assignment.
- **Missing startle responses (0.5%) were excluded. How did the authors define missing startle responses?** Missing startle responses were caused by recording artifacts, which is now clarified in lines 542-543.

Minor comments:

1. Parts of the introduction, especially with regard to the details of the molecular pathways, do not seem to be essential for the rationale of the experiments. Perhaps these parts could be moved to the discussion section and the introduction could be shortened a bit. Although the details of the molecular pathways are indeed not essential for the rationale of the experiments, the neurobiological processes that are allegedly targeted by propranolol are important to mention in order to prevent misunderstandings on the conceptualization of the reconsolidation intervention (e.g., see for instance a recent review paper where it was suggested that propranolol in our reconsolidation studies works as a standard anxiolytic drug (Steenen et al 2016., Propranolol for the treatment of anxiety disorders: Systematic review and meta-analysis, *Journal of Psychopharmacology*, 30, 128-139). However, in our previous publications we have extensively demonstrated that the fear-reducing effects are specific for the reactivated memory trace, depend on prediction error etc, which cannot be reconciled with traditional pharmacological interventions such as anxiolytic drugs. In view of these potential misunderstandings, we believe that it is still important to explain the underlying neurobiology of the reconsolidation intervention.

2. It would be helpful to spell out β -AR in the title for non-experts. Also, the abbreviation PE in extended Figures 1, 2 and 3 should be defined. The title of the manuscript has been changed now. PE driven learning is now explained in the legend of Extended Data Fig. 1 - see lines 782-784, Extended Data Fig. 2 - see lines 799-800, Extended Data Fig 3. - see line 817.

3. Line 136: The abbreviation CS1-R is not defined here. Could it simply be termed CS1 here? We now clarify the abbreviation CS1-R in lines 119-120

4. Line 190: after reactivation instead of retrieval? We have replaced “retrieval” with reactivation throughout the manuscript.

5. Typos: line 305 “This in”, line 322 “took play”, line 364 “a ground references”, line 709 “extinction trial group”. We have addressed the typos as well.

Reviewers' comments:

Reviewer #1 (Remarks to the Author):

Kindt and Soeter followed my suggestions and revised the manuscript accordingly. Only a few minor comments remain.

-The authors now report the gender distribution within each of the samples, but do not report if there is any inequality in gender distribution between groups (as tested for age, fear assessments, US evaluation and shock intensity, see line 761).

-The discussion should include a sentence that (I think in accordance with previous studies) the pharmacological induced amnesia affected startle responses, but not US expectancy.

-Startle responses were defined as "Peak amplitudes [...] over the period of 50 - 150 ms following probe onset" (line 535). Just to get it right: Does that mean that a baseline was subtracted from the maximum value? Additionally, the unit of these responses (e.g. microvolts) is missing at each of the figure axis. Related to the startle measurements, what was the reasoning against T-transformation of startle responses (other than using this procedure in previous experiments)? Recommendations for the analysis of startle responses suggest these transformations (Blumenthal *Psychophysiology*, 42 (2005)): "For reasons as yet largely unknown, wide individual differences in absolute blink magnitude are observed, and this variation is often unrelated to the experimental phenomena of interest. [...] For this reason, many experimenters standardize blink magnitudes in some way, such as using all blinks for a given subject as the reference distribution and reporting the results as z or T (mean 50, SD 10) scores." The authors should consider to state their reasoning against T-transformation in the manuscript.

-The authors now report the criteria for rejecting a single trials of Startle responses, but was there a criteria to reject a participant (e.g. less than 2/3 of valid responses)? And if so, how many participants were excluded?

Reviewer #2 (Remarks to the Author):

The authors satisfactorily addressed most of my concerns and made appropriate changes in the manuscript. However, I have one remaining concern with regard to my previous comment #1. The authors argue that β -ARs are only involved during a limited time window of 2-3 hours after reactivation. However, even with the extended discussion provided on line 331-349, I am not convinced that the data actually support this conclusion. Specifically, the authors argue that "if early β -AR activity before the 2-3 h time window had been responsible for the post-reactivation amnesia, then the administration of propranolol 1 h following memory reactivation would have missed this window." (line 334-337). This statement is only correct if the authors assume that a potential early window would be the only critical time window (e.g. 0-2 hours). In this case, the administration of propranolol 1 hour post-reactivation would have indeed missed this window. However, it is also possible that the critical time window is wider than postulated by the authors, e.g. 0-3 hour or 1-3 hour post-reactivation. Such a time window would be consistent with all of the presented

data, i.e. an amnesic effect of propranolol administration 1 hour before, right after as well as 1 hour after reactivation. I agree that the present data clearly show that the 2-3 hour window is critical. But the data do not exclude that the 0-2 hour interval is also critical. That is, the authors cannot exclude that propranolol administered 1 hour before reactivation acted 0-1 hour after reactivation and propranolol administered right after reactivation acted 1-2 hour post-reactivation, while propranolol administered 1 hour after reactivation acted 2-3 hour post-reactivation. I am not saying that the postulated 2-3 hour window is wrong; future studies may show that the authors' hypothesis of such a narrow time window is correct. However, the present data cannot conclusively disprove a wider 0-3 h or 1-3 h time window. This should at least be discussed.

Reviewer #1 (Remarks to the Author):

Kindt and Soeter followed my suggestions and revised the manuscript accordingly. Only a few minor comments remain.

1 - The authors now report the gender distribution within each of the samples, but do not report if there is any inequality in gender distribution between groups (as tested for age, fear assessments, US evaluation and shock intensity, see line 761). Following the suggestion of the reviewer, we now report the rather equal gender distribution among the groups in Supplementary table 1. Again, present sample sizes are too small to properly test for gender effects.

2 - The discussion should include a sentence that (I think in accordance with previous studies) the pharmacological induced amnesia affected startle responses, but not US expectancy. We have now stated in the discussion section that the pharmacological manipulation of memory reconsolidation affected the fear potentiated startle response. Given that the US expectancies are only reported in the Supplementary information, we leave the statement of “Propranolol does not affect the US expectancy ratings” to the Supplement (see Fig. 1, Fig. 2, and Fig. 3).

3 - Startle responses were defined as “Peak amplitudes [...] over the period of 50 - 150 ms following probe onset” (line 535). Just to get it right: Does that mean that a baseline was subtracted from the maximum value? Additionally, the unit of these responses (e.g. microvolts) is missing at each of the figure axis. Related to the startle measurements, what was the reasoning against T-transformation of startle responses (other than using this procedure in previous experiments)? Recommendations for the analysis of startle responses suggest these transformations (Blumenthal Psychophysiology, 42 (2005)):” For reasons as yet largely unknown, wide individual differences in absolute blink magnitude are observed, and this variation is often unrelated to the experimental phenomena of interest. [...] For this reason, many experimenters standardize blink magnitudes in some way, such as using all blinks for a given subject as the reference distribution and reporting the results as z or T (mean 50, SD 10) scores.” The authors should consider to state their reasoning against T-transformation in the manuscript. Startle responses were indeed baseline corrected. For reasons of clarification, it is now stated that “Baseline-to-peak amplitudes [...] over the period of 50-150 ms following probe onset” – see line 565. Startle amplitudes are indeed expressed in microvolts. We now display this at each of the figure axis. We entirely agree with the reviewer that wide individual differences in absolute blink magnitude are observed (e.g. Blumenthal et al. 2005). Given that we (1) correct the peak amplitudes for baseline and (2) do not investigate single trials but rather compare the differential responding (e.g. CS1 vs. CS2), we already control for individual differences. Furthermore, we prefer to use the raw data – especially since no preferred method for standardization has emerged yet (see also Blumenthal et al. 2005, page 11).

4 - The authors now report the criteria for rejecting a single trials of Startle responses, but was there a criteria to reject a participant (e.g. less then 2/3 of valid responses)? And if so, how many participants were excluded? Upon inspection of the data, we did not notice any participants who failed to exhibit a large number of startle responses and therefore did not use any specific criteria for

“nonresponders”. The maximum of invalid responses was 7 % (i.e., 4 of the 57 trials), so we didn’t reject any participant from the dataset. This has now been stated in the methods on line 574-575.

Reviewer #2 (Remarks to the Author):

1 - The authors satisfactorily addressed most of my concerns and made appropriate changes in the manuscript. However, I have one remaining concern with regard to my previous comment #1. The authors argue that β -ARs are only involved during a limited time window of 2-3 hours after reactivation. However, even with the extended discussion provided on line 331-349, I am not convinced that the data actually support this conclusion. Specifically, the authors argue that “if early β -AR activity before the 2-3 h time window had been responsible for the post-reactivation amnesia, then the administration of propranolol 1 h following memory reactivation would have missed this window.” (line 334-337). This statement is only correct if the authors assume that a potential early window would be the only critical time window (e.g. 0-2 hours). In this case, the administration of propranolol 1 hour post-reactivation would have indeed missed this window. However, it is also possible that the critical time window is wider than postulated by the authors, e.g. 0-3 hour or 1-3 hour post-reactivation. Such a time window would be consistent with all of the presented data, i.e. an amnesic effect of propranolol administration 1 hour before, right after as well as 1 hour after reactivation. I agree that the present data clearly show that the 2-3 hour window is critical. But the data do not exclude that the 0-2 hour interval is also critical. That is, the authors cannot exclude that propranolol administered 1 hour before reactivation acted 0-1 hour after reactivation and propranolol administered right after reactivation acted 1-2 hour post-reactivation, while propranolol administered 1 hour after reactivation acted 2-3 hour post-reactivation. I am not saying that the postulated 2-3 hour window is wrong; future studies may show that the authors’ hypothesis of such a narrow time window is correct. However, the present data cannot conclusively disprove a wider 0-3 h or 1-3 h time window. This should at least be discussed.

We agree with the reviewer that on basis of the present results we cannot exclude the possibility of more than one peak of β -AR activity post reactivation. Indeed, the administration of propranolol 1 h before memory reactivation could potentially has blocked an earlier peak of β -AR activity, whereas administration of propranolol 1 h post reactivation could block a later peak (2-3 h). We have toned down our conclusion and changed our text accordingly (see Discussion in blue).